# The Resilient Teacher: Unveiling the Positive Impact of the Collaborative Practicum Model on Novice Teachers

**Yonit Nissim** [1],* **and Alexandra Danial-Saad** [2] 

1 Department of Education and Learning, Tel Hai College, Kiryat Shemona 1220800, Israel
2 Department of Occupational Therapy, University of Haifa, Haifa 3103301, Israel; asaad@univ.haifa.ac.il
* Correspondence: yonitn@telhai.ac.il

**Abstract:** The present quantitative, non-experimental comparative study, delves into the long-term effects of the collaborative practicum model (specifically the "academy-class" model) on novice teachers. The research aims to discern disparities in the professional self-efficacy of novice educators who underwent training within the collaborative practicum model as opposed to those who adhered to the conventional teaching model. This comparative analysis is based on three variables: perception of the teaching profession, professional self-efficacy, and socio-economic security. Furthermore, the study examines whether the collaborative model contributes to cultivating more favorable attitudes toward the teaching profession and a greater inclination to continue teaching for an extended period exceeding three years. The study encompasses a cohort of 436 Bachelor of Education (B.Ed.) graduates from 22 Israeli Higher Education Institutions who completed their degrees within the past five years. The research findings underscore a higher level of teaching efficacy, socio-economic security, and a more positive outlook among those who participated in the collaborative practicum and expressed their intent to persist in the teaching profession. These outcomes underscore the vital role of the collaborative practicum model, hinting at its potential to positively influence the retention rate within the teaching profession. Furthermore, it underscores the crucial connection between comprehensive and meaningful training within a collaborative practicum framework and the sustainable professional growth of educators. This robust training approach can potentially secure the continued presence of dedicated and enthusiastic educators in the field over the long term.

**Keywords:** practicum; teacher training; collaborative model; academy-class

## 1. Introduction

In recent years, a number of studies have examined the effect of the practicum on Preservice Teachers (PSTs) during their studies or internship in Israel [1–5]. The studies found differences between the collaborative and the traditional models on PST training. However, there is still a lack of in-depth research on the long term effect of the collaborative model on novice teachers, who either persevered, or alternatively chose to drop out from the profession [6]. This study aims to address this existing gap in the research.

## 2. Literature Review

### 2.1. Perceptions of the Role of the Teacher among Preservice Teachers

Preservice Teachers (PSTs) approach their professional training with different perceptions and images of the essence of teaching, the desired interaction with students, and their self-image as teachers rooted in their childhood [7,8]. In addition, [9] argue that the teachers' perception of "self" represents the way of thinking and behavior and involves reciprocal relations between the teachers and themselves (status, image, roles, and past experience) and between them and their environment, such as the attitude of the principal, teaching colleagues, students and their parents [10] argue that these factors influence how

the teachers attribute their importance positively or negatively. Furthermore, these perceptions have an impact on their professional implementation as reflected in their professional work [11].

Preservice Teachers (PSTs) at the onset of their training can be a challenging endeavor. This is because the new knowledge they acquire through their courses often tends to validate or reinforce their existing preconceptions and notions without effecting significant changes [12]. However, an opposing viewpoint suggests that it is indeed possible to transform attitudes and reshape the deeply ingrained preconceptions of PSTs through hands-on practical experience in real-world educational settings. This practical experience has been shown to exert a substantial influence on their attitudes and decisions regarding their future careers [13]. Recent research further underscores the pivotal role of teacher training processes, particularly the practicum, in serving as a constructive element for shaping their attitudes [2–5].

### 2.2. Transition from Preservice Teacher to Novice Teacher

Novice teachers often find themselves grappling with a range of challenges that have been vividly characterized as "shocks"—encompassing aspects like reality shock, culture shock, live classroom shock, and first-year shock—alongside a relentless "struggle for survival" [14]. These descriptions poignantly paint a picture of stress, difficulties, inadequate functioning, feelings of frustration, isolation, and loss of self-confidence [15–18]. In addition to the immense mental and emotional burden they shoulder, novice teachers may often find themselves lacking the requisite self-efficacy to effectively cope with these difficulties [19]. This dearth of self-efficacy stems from several factors, including their limited practical knowledge, inadequate classroom management skills, emotional challenges in dealing with the rigors of the profession, placements in particularly challenging classroom environments, insufficient support within school settings, and a lack of robust mentoring and socialization at the organizational level of the school [20]. So, the transition period for these novice teachers can be a rollercoaster of emotions, ranging from intense anxiety to exhilaration, and the outcomes of these initial difficulties frequently manifest as alarmingly high attrition rates among novice teachers, estimated at approximately 30% to 50%, and subsequently contribute to teacher shortages [21,22]. As per a report from Israel's Central Bureau of Statistics, a substantial 19.7% of newly qualified teachers exit the profession within their first three years [23].

In summary, their self-efficacy, perceptions, and overall attitudes towards the teaching profession are fundamentally shaped by their training experiences [24]. This crucial aspect warrants further exploration and in-depth examination, which we will now delve into.

### 2.3. Self-Efficacy in Teaching

The concept of "sense of self-efficacy in teaching" is intricately tied to the socio-cognitive learning theory, as put forth by [25]. Within this framework, a teacher's sense of self-efficacy is specifically defined as the level of belief in one's capacity to effectively organize and execute the actions required to attain desired professional outcomes within the classroom setting of a school or kindergarten [26–28] further contend that self-efficacy should encompass a teacher's ability to cultivate positive interpersonal relationships with individuals within the school organization, particularly with key figures such as the principal, and to exert an influence over what happens in the school. In light of this, teachers endowed with a robust sense of self-efficacy are better positioned to impart knowledge, competencies, and skills to their students [29].

Furthermore, those teachers who effectively carry out their responsibilities and achieve the desired outcomes tend to experience a deep sense of satisfaction and harbor positive sentiments towards their chosen profession [30]. Consequently, it has been substantiated that Preservice Teachers (PSTs) undergo a noticeable enhancement in their sense of self-efficacy over the teaching training processes. Their practical experiences during the practicum, coupled with the knowledge they acquire, bear a substantial impact not only

on how they perceive their own abilities but, perhaps even more crucially, on how they apply the knowledge and skills they have gained during their practicum [31].

*2.4. The Role of the Practicum in Teacher Training*

The journey towards becoming an educated and professionalized teacher is a multifaceted and intricate process. It melds theoretical instruction from academic institutions with hands-on clinical training, a practicum the real world of educational settings like schools and kindergartens. Throughout this journey, prospective teachers undergo a process of self-discovery, refining their knowledge and skills, while simultaneously shaping their attitudes, identities, educational perspectives, and critical thinking abilities [32,33].

The practicum experience is the most significant stage in PST training, and in recent years, we have witnessed it becoming the highlight of teacher education programs [34]. The practicum provides PSTs with a unique opportunity for direct immersion within the school environment, enabling them to acquaint themselves with all its facets, interact with their peers, and understand their roles. It is a shared professional and personal journey between PSTs and their mentors [32,35,36]. The positive impact of the practicum on PSTs is particularly pronounced during their transition from theoretical academic study to active teaching in the classroom and full engagement within the school environment. It plays a pivotal role in shaping their attitudes towards their future professional roles as teachers, highlighting the paramount importance of the practicum in fostering an optimal learning experience. Furthermore, it serves as a facilitating and supporting element in the progression from PST to novice teacher [16,37,38].

In the realm of teacher training in Israel, two primary models of the practicum are employed:

1.  The traditional model—in which students are assigned to schools for a practicum of one day (six hours) a week, accompanied by a pedagogical instructor [39].
2.  The collaborative model named "Academy-Class" (similar to the model professional development school [PDS]). The PST undergoes varied intensive experiences with high involvement in school life, while receiving mediation and feedback from the pedagogical instructor and from the mentoring teacher [16,40]. The PST gradually integrates into teaching work, beginning with observing, assisting and one-on-one teaching, up to full co-teaching with a coaching teacher [41]. The collaborative practicum model experience contributes to the improvement in teacher training and to the professional development of the coaching teachers [41–44].

*2.5. Advantages of the Collaborative Practicum Model (PDS and Academy-Class)*

Various studies underscore the profound impact of extensive exposure to teaching experiences, particularly when paired with a teacher mentor and pedagogical instructor, in preparing PSTs for their future roles as teachers [4,16,45]. Research findings reveal that graduates of the collaborative model exhibit a remarkably high sense of self-efficacy and readiness for the teaching profession, significantly enhancing their prospects of successful integration into the teaching workforce [46,47]. The collaborative practicum model, as exemplified within the Academy-class program, holds a distinct advantage in facilitating a smooth transition into the internship year and a higher likelihood of successful entry into the teaching profession compared to students trained in the traditional practicum model [46]. In the collaborative academy-class practicum model, PSTs undergo a transformation similar to that of novice teachers, cultivating their dedication to teaching and the art of pedagogy. This experience effectively fortifies their preparation for the teaching profession, notably influencing their seamless integration into the realm of teaching [5]. The PSTs participating in the collaborative practicum model consistently acknowledge its substantial role in their training journey, acknowledging its significant contribution to their readiness to embark on their teaching careers [2,19].

Moreover, a recent study conducted by [1] delves into the differences between practicum models among PSTs, illuminating significant disparities favoring the collabora-

tive Academy-class practicum model. These differences encompass various facets, including content knowledge, teaching methods, adaptability to students, parent engagement, and pedagogical benefits such as the acquisition of effective work habits and enhanced collegial relationships within the school staff [1].

Despite the abundance of research exploring the impact of the collaborative model on multiple facets of teacher training, there remains a notable gap in understanding the long-term effects of different practicum models on novice teachers' sense of self-efficacy and their aspirations to remain within the teaching profession. This study endeavors to evaluate the extent to which the collaborative practicum model influences the sense of self-efficacy among novice teachers. It further seeks to investigate whether the type of practicum (collaborative vs. traditional) has a lasting impact on their self-efficacy and influences their commitment to continued engagement in teaching. These assessments will rely on the reports and experiences of novice teachers who completed their teacher training within the previous five years.

### 3. Methodology

The present research constitutes a quantitative, non-experimental comparative study, aimed at assessing the enduring impact of the collaborative model of the practicum on the professional self-efficacy of recent graduates who completed their teacher training within the previous five years, as juxtaposed with graduates who underwent the traditional practicum model. Thus, the following hypotheses were suggested:

(A) Comparative analysis: there is a difference in research measures (self-efficacy in teaching; socioeconomic security; educational impact on next generation, professional self-realization and academy-field collaboration) between PST in the collaborative practicum model in contrast to PST in the traditional model. This research undertakes a meticulous comparison of various measurable aspects associated with the collaborative practicum model (research group) in contrast to the traditional model (control group), specifically focusing on self-efficacy in teaching and attitudes towards the teaching profession. This comparative analysis is conducted alongside a control group comprised of teachers who have undergone the traditional practicum model.

(B) Exploration of Future Commitment: There is a positive correlation between research measures and remaining in teaching (more than 3 years). This study delves into the influence of the collaborative model on the inclinations of novice teachers regarding their sustained engagement within the education system. It seeks to ascertain whether the collaborative model contributes to a higher likelihood of novice teachers remaining in the teaching profession for an extended period (i.e., beyond three years), or conversely, if it has an impact on their attrition from the profession. This exploration also encompasses individuals who may have already withdrawn from the profession at the time of the study.

(C) Interplay of factors: There is a difference in research measures regarding the intention to remain in teaching between PST in the collaborative practicum model in contrast to PST in the traditional model. By discerning the relationships between these variables, the study aims to provide a holistic understanding of the multifaceted impact of the collaborative model on the attitudes and intentions of novice teachers within the educational landscape.

#### 3.1. Research Population

Participants were 436 graduates from all academic teacher education institutions in Israel who had completed their preservice training in the previous five years. The sample included a research group, graduates who had participated in the academy-class program (N = 309) and a control group of those who had not (N = 127). Of the 436 participants, 365 were women (83.7%) and 71 were men (16.3%) aged 21–60 (mean age = 30.16). Most respondents were married (56.8%), about half were Jews (50.1%), and about one third were Muslims (37.4%), and the rest were either Druze (7.2%) or Christians (5.3%).

### 3.2. Research Method

The study obtained ethical approval from the ethics committee of the Research Authority at Tel Hai Academic College to ensure its compliance with ethical standards.

This quantitative research employed a well-validated questionnaire designed to assess several key dimensions. The questionnaire addressed self-efficacy, adapted from the work of [48] and tailored to the specific context of teaching. Additionally, it explored attitudes towards the teaching profession, based on the framework established by [49] Katzir et al. (2004), and pedagogical applications, derived from a national test administered by the Ministry of Education. Finally, the Inclination to Sustain a Career in the Teaching Profession, categorized as either short-term (less than three years) or long-term (more than three years), served as a measure of retention. To assess these measures, participants were asked to indicate their level of agreement with 26 statements on a Likert-like scale, ranging from 1 (not at all) to 5 (very much so). The pedagogical application measure encompassed aspects such as the adaptability to student diversity, evaluation skills, and attitudes regarding the teaching profession, pedagogical practices, and collaboration with fellow teaching colleagues. The questionnaire was accompanied by a brief explanation regarding the study's objectives. Notably, the research tool had been employed in a previous study involving a similar population, and all categories within the questionnaire had undergone thorough validation and reliability testing by fellow researchers (Sassoon et al., 2020, p. 461). Collaboration with the original authors of the questionnaire was conducted to further validate the instrument. A face validation procedure was then executed, involving four content experts holding Ph.D. degrees in education, in accordance with the established literature [50].

The distribution of the questionnaire was executed by sending it to an email or WhatsApp. The sample frame: a list provided by the Ministry of Education, reaching approximately 8000 recent graduates in the teaching profession, who were invited to access the questionnaire via a Google Drive link with ensuring the participants' anonymity.

Data analysis was performed using the IBM SPSS version 25-software package. The subsequent section presents descriptive demographic data regarding the participants, followed by an overview of the primary research measures. The study then proceeds to explore correlations and differences, with a focus on potential relationships between the research variables. Research questions were assessed through independent *t*-tests, ANOVA, and chi-squared tests of independence.

The study population: N = 435. Table 1 is describing the distribution of respondents by demographic variables.

**Table 1.** The distribution of respondents by demographic variables.

|  | N | % | Min | Max | M | SD |
|---|---|---|---|---|---|---|
| Age |  |  | 21.0 | 60.0 | 30.16 | 7.85 |
| Gender |  |  |  |  |  |  |
| Female | 365 | 83.7 |  |  |  |  |
| Male | 71 | 16.3 |  |  |  |  |
| Assigned a teaching position |  |  |  |  |  |  |
| Yes | 69.8 | 70.6 |  |  |  |  |
| No | 29.1 | 29.4 |  |  |  |  |
| Intention to work in teaching |  |  |  |  |  |  |
| Less than 3 years | 71 | 16.2 |  |  |  |  |
| More than three years | 366 | 83.8 |  |  |  |  |
| Participation in the academy-class collaborative practicum |  |  |  |  |  |  |
| Yes | 309 | 70.9 |  |  |  |  |
| No | 127 | 29.1 |  |  |  |  |

## 4. Findings

A comparison between the distribution by the collaborative model practicum (research group) and the traditional model (control group), in terms of integration into teaching, was carried out, presented in Table 2.

**Table 2.** The distribution by integration into teaching in a comparison between the collaborative model practicum and the traditional model. (N = 435).

| | | Integration into Teaching | | |
|---|---|---|---|---|
| **Model of Practicum** | | **Integration into Teaching** | **Did Not Integrate** | **Total** |
| Traditional practicum | N | 88 | 38 | 126 |
| | % | 69.8% | 30.2% | 100.0% |
| Collaborative practicum | N | 219 | 90 | 309 |
| | % | 70.9% | 29.1% | 100.0% |
| Total | N | 307 | 128 | 435 |
| | % | 70.6% | 29.4% | 100.0% |

As shown in Table 2, 70.9% of the participants in the collaborative practicum became involved in teaching immediately after graduation, while 69.8% of those who participated in a traditional practicum integrated into teaching immediately after graduation. In other words, a similar percentage of participants were integrated into teaching, both the research and the control group. A chi-squared test determined that the differences were not significant: $x - 2(1) = 0.05$, $p > 0.05$.

As shown in Table 3, 81.6% of participants in the collaborative practicum intend to remain in teaching for more than 3 years, while 89.0% of participants in the traditional practicum intend to stay. That is, a higher percentage of participants in the traditional practicum intend to remain in teaching. A chi-squared test determined that the differences were borderline significant (with a 10% significance level): $x^2(1) = 3.64$, $p = 0.056$.

**Table 3.** Distribution by intention to remain in teaching according to practicum model participation (N = 436).

| | | Intention To Remain in Teaching | | |
|---|---|---|---|---|
| **Practicum Model** | | **More than 3 Years** | **Less than 3 Years** | **Total** |
| Traditional model | N | 113 | 14 | 127 |
| | % | 89.0% | 11.0% | 100.0% |
| Collaborative model | N | 252 | 57 | 309 |
| | % | 81.6% | 18.4% | 100.0% |
| Total | N | 365 | 71 | 436 |
| | % | 83.7% | 16.3% | 100.0% |

*t*-tests were performed for independent samples to examine the differences in research measures between participants in the collaborative practicum and those who participated in the traditional practicum. Table 4 below shows the averages for both groups and the test results.

As seen from the findings for all measures, the averages of participants in the collaborative model were higher than the averages of the participants in the traditional model. In addition, significant differences were found for the measures of socio-economic security: $t(434) = 2.37$, $p < 0.05$; professional self-realization: $t(434) = 2.91$, $p < 0.01$; and in the academy–field partnership: $t(432) = 4.36$, $p < 0.05$.

*t*-tests were performed for additional independent samples to examine the association in the study measures between participants who intend to work in teaching for more than three years and those who intend to work in teaching for up to three years, Table 5 below shows the averages in both groups and the test results.

**Table 4.** Differences in research measures by *t*-test results of collaborative practicum participation traditional model (N = 436).

| Measure | Traditional Model (N = 127) | | Collaborative Model (N = 309) | | |
|---|---|---|---|---|---|
| | M | SD | M | SD | T |
| Sense of efficacy in teaching | 4.3 | 0.65 | 4.4 | 0.7 | 1.39 |
| Socio-economic security | 3.52 | 1.06 | 3.77 | 1 | 2.37 * |
| Educational impact on the next generation | 4.6 | 0.59 | 4.66 | 0.55 | 1 |
| Professional self-realization | 4.16 | 0.76 | 4.38 | 0.69 | 2.91 ** |
| Academy-field collaboration | 2.79 | 1.36 | 3.4 | 1.3 | 4.36 ** |

\* $p < 0.05$ ** $p < 0.01$.

**Table 5.** Correlation between research indices by intention to remain in teaching and *t*-test results (N = 437).

| Index | Less than 3 Years (N = 71) | | More than 3 Years (N = 366) | | |
|---|---|---|---|---|---|
| | M | SD | M | SD | T |
| Sense of efficacy in teaching | 4.13 | 1.00 | 4.42 | 0.59 | 2.32 * |
| Socio-economic security | 3.63 | 1.24 | 3.72 | 0.98 | 0.56 |
| Educational impact on the next generation | 4.47 | 0.80 | 4.67 | 0.50 | 2.03 * |
| Professional self-realization | 4.11 | 0.97 | 4.36 | 0.65 | 2.10 * |
| Academy-field collaboration | 3.25 | 1.35 | 3.21 | 1.34 | 0.19 |

\* $p < 0.05$.

The sense of efficacy in teaching of those who intend to remain in teaching for more than 3 years was found to be significantly greater than that of those who intend to remain in teaching for less than three years: t(78) = 2.32, $p < 0.05$. Significant differences were also found for the measures of educational impact on the next generation (t(81) = 2.03, $p < 0.05$), and professional self-realization (t(83) = 2.10, $p < 0.05$). The hypothesis was confirmed, since the stronger the sense of efficacy and the more positive the attitudes, the greater the intention to remain in teaching.

To examine differences between all study measures and groups, F variance tests were performed for additional independent samples. Table 6 below presents the averages within each group, the intention to remain in teaching, and the research measures, and presents the results of the variance tests.

**Table 6.** Differences in research measures according to intention to remain in teaching and participation in the collaborative and traditional practicum model and variance test results (N = 436).

| Index | Less than 3 Years Traditional Model (N = 14〉 | | More than 3 Years Traditional Model (N = 113) | | Less than 3 Years Collaborative Model (N = 57) | | More than 3 Years Collaborative Model (N = 252) | | |
|---|---|---|---|---|---|---|---|---|---|
| | M | SD | M | SD | M | SD | M | SD | F |
| Sense of efficacy in teaching | 4.07 | 1.01 | 4.33 | 0.59 | 4.15 | 1.01 | 4.46 | 0.59 | 4.50 ** |
| Socio-economic security | 4.64 | 0.56 | 4.59 | 0.6 | 4.43 | 0.84 | 4.71 | 0.45 | 4.15 ** |
| Educational impact on next generation | 3.96 | 1.09 | 4.19 | 0.71 | 4.14 | 0.95 | 4.43 | 0.61 | 5.92 ** |

\** $p < 0.01$.

Table 6 shows the differences in the sense of self-efficacy in comparison between those who intend to remain in teaching (who participated in the collaborative practicum model) and those who do not intend to remain in teaching (and who did not participate in the collaborative model). These differences were significant in the variance tests: F(3,432) = 4.50, $p < 0.01$.

The highest sense of socio-economic security was found among those who intend to remain in teaching and participated in the collaborative model (4.71), followed by those who do not intend to remain in teaching and who did not participate in the collaborative model (4.64), and those who intend to remain in teaching but did not participate in the collaborative model (4.59). The sense of socio-economic security of those who do not intend to remain in teaching and who participated in the collaborative model (4.43) was the lowest. These differences were found to be significant in the variance tests: $F(3,432) = 4.15$, $p < 0.01$.

For those who intend to remain in teaching and who participated in the collaborative practicum, the educational impact on the next generation is the greatest (4.43), followed by those who intend to remain in teaching but participated in the traditional model (4.19). Next were those who do not intend to remain in teaching and participated in the collaborative practicum (4.14). The educational impact factor among those who do not intend to remain in teaching and who participated in the traditional model (3.96) was found to be the lowest. These differences were significant in the variance tests: $F(3,432) = 5.92$, $p < 0.01$.

Hence, the averages of for those who intend to remain in teaching and who participated in the collaborative model were the highest for each of the three measures.

## 5. Discussion

The study's main findings indicate that the collaborative practicum model produces more "positive" and resilient teachers. it also indicate a positive correlation between the sense of self-efficacy and the intention to persevere in the teaching profession. This has a significant long-term effect on their retention in the system, on their desire to remain in teaching and on their motivation. These findings are consistent with the goals and assumptions of the collaborative "academy-class" practicum model, which seeks to improve the quality of teachers' training and professional development, as well as strengthen the status of teaching as a profession [41]. Moreover, the current study findings support and expand previous studies on the success of the collaborative practicum model and found that its graduates had high self-efficacy and preparedness for their role as teachers, leading to improved integration into the teaching profession. These outcomes are in line with studies by [4,46,47] with recent research even reporting significant differences in favor of self-efficacy levels of PSTs trained via the collaborative model [1]. These cumulative findings further emphasize the efficacy of the collaborative model in nurturing well-prepared and confident educators.

## 6. Conclusions

The central research hypothesis posits that the collaborative practicum model exerts a positive influence on the self-efficacy of novice teachers, thereby enhancing their confidence and competence in their teaching roles. The research hypothesis was confirmed. As seen from the main finding, the current study indicates that a high percentage (81.6%) of participants from the collaborative practicum model express an intention to remain in teaching profession for more than three years. These findings are in line with previous studies indicating that the intensive practicum in the collaborative model helps PSTs enhance their understanding of the tension between theory and practice, enables their gradual integration into the field of education, and creates a supportive environment for them to learn [5,51]. A study by [52] also indicated that teachers' self-efficacy level is a significant factor influencing teacher effectiveness and degree of satisfaction with the profession and thus their retention or dropout [53] also highlight the importance of self-efficacy in the study on its impact on teaching practices and teachers' interpersonal and environmental factors.

The findings of this study also reveal a notable positive correlation between the sense of self-efficacy and the determination to persist in the teaching profession. These insights hold substantial significance as they underscore the critical need for improvements in both preservice teacher training and in-service professional development. Enhancing and sustaining motivation and the desire to continue in the teaching profession are pivotal objec-

tives, particularly in light of the mounting data concerning teacher attrition in Israel [41]. Addressing this issue necessitates the implementation of comprehensive measures aimed at supporting and retaining educators.

In summary, the main conclusions drawn from the current research underscore the substantial contribution of the collaborative practicum model within the landscape of teacher training. This model serves as a pivotal component in nurturing attributes such as optimism, mental resilience, self-confidence, and adaptive attitudes among prospective teachers. Equipped with these qualities, educators are better prepared to maintain their dedication to the profession, subsequently enabling them to make more effective contributions to their students and society as a whole, thus fulfilling their roles as educational leaders.

### 6.1. Study Contribution

The distinctiveness of this study lies in its capacity to offer fresh practical insights within the domain of pre-service teacher training, with a specific focus on the relatively unexplored landscape of the collaborative practicum approach. To our knowledge, no comprehensive, large-scale inquiry has ventured into this critical territory, rendering this research an innovative endeavor. Furthermore, this study, while making a modest contribution, assumes a significant role in both practical and theoretical dimensions of the field.

The analytical approach employed in this study carries the potential to provide an initial glimpse into the prospective trajectory of aspiring educators in terms of their commitment to the educational and teaching professions. It, in effect, offers valuable resources, encompassing tools and solutions directed at augmenting pre-service teacher training and enhancing practical experiences in the field. This insightful perspective is poised to facilitate well-informed decision making and the strategic implementation of measures geared towards fortifying teacher retention, thereby nurturing a more resilient and highly motivated teaching workforce.

### 6.2. Research Limitations

This study specifically concentrated on the collaborative practicum model within the broader spectrum of teacher training processes, making it imperative to exercise caution when generalizing its findings to encompass all teacher training programs. Furthermore, the participants involved in this research represent novice teachers who concluded their training within the previous five years. It is conceivable that the ongoing in-service professional development they undergo may exert an influence on their perceptions concerning the likelihood of persisting in the profession or considering discontinuation in the future.

In light of the study's limitations, there are several recommendations for further research that warrant exploration. Firstly, it would be advantageous to re-examine the same research population approximately five years down the line to assess whether the anticipated trends regarding persistence or dropout have indeed materialized. Additionally, different subgroups of novice teachers, characterized by varying sociodemographic, cultural, and regional attributes, should be scrutinized to ascertain how attitudes and factors influencing persistence might differ among these diverse cohorts.

**Author Contributions:** The co-authors wrote the article in equal contribution. All authors have read and agreed to the published version of the manuscript.

**Funding:** This research was funded by the research's authorities of the Tel Hai College & University of Haifa.

**Institutional Review Board Statement:** The study was conducted in accordance with the Declaration of Helsinki, and approved by the Institutional Ethics Committee of the Ohalo College No 10520 (protocol code: 07112020EC; approval data: 7 November 2020).

**Informed Consent Statement:** Not applicable.

**Data Availability Statement:** For data Availability and farther information please contact the authors: yonitn@telhai.ac.il; asaad@univ.haifa.ac.il.

**Conflicts of Interest:** The authors declare no conflict of interest.

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
