# Peer review of "The Resilient Teacher: Unveiling the Positive Impact of the Collaborative Practicum Model on Novice Teachers"

_education, doi:10.3390/educsci13111162_

Round 1

Reviewer 1 Report

Comments and Suggestions for Authors

This paper adequately described comparisons between the professional self-efficacy of novice teachers who were either trained in the collaborative practicum model or experienced the traditional model. The models were well explained, and the data differences showed how the collaborative practicum model offered PSTs a superior experience and preparedness for the profession.

  • General concept comments
    Article: highlighting areas of weakness, the testability of the hypothesis, methodological inaccuracies, missing controls, etc.

Whilst as noted this is a quantitative, comparative study it would have been nice to include some personal insights from the PSTs. The questions may have elicited written responses, if so, some of these could be used to enlighten the reader on PST experiences. Otherwise the data presented was sound, clearly itemised and presented

  • The manuscript is clear, relevant for the field and presented in a well-structured manner 
  • The cited references are relevant
  • The manuscript’s results are reproducible and based on the details given in the methods section
  • The figures/tables/images/schemes are appropriate and are easy to interpret and understand
  • The conclusions are consistent with the evidence and arguments presented
  •  The ethics statement is included  
Comments on the Quality of English Language

There is some scope for the article to be edited and checked for English grammar and syntax 

Author Response

Dear reviewer,

First, we would like to thank you wholeheartedly for your eye-opening comments. These have been tremendously helpful in guiding the major changes we have made to this paper. We have made significant changes based on all your comments as you can see in "Track the changes". We would like to emphasize the following points that have been substantially modified in line with your comments.

All your comments have been taken into account and the article has been modified accordingly.  

I will answer each comment separately. For convenience, I have attached your comments) in a bold font( and answered ,below in red ,each comment in accordance with what appears in the corrections made in the body of the article.

This paper adequately described comparisons between the professional self-efficacy of novice teachers who were either trained in the collaborative practicum model or experienced the traditional model. The models were well explained, and the data differences showed how the collaborative practicum model offered PSTs a superior experience and preparedness for the profession.

  • General concept comments
    Article: highlighting areas of weakness, the testability of the hypothesis, methodological inaccuracies, missing controls, etc.

Thanks for the valuable comments. We significantly corrected these parts, as you can see in the " Track the changes"

Whilst as noted this is a quantitative, comparative study it would have been nice to include some personal insights from the PSTs. The questions may have elicited written responses, if so, some of these could be used to enlighten the reader on PST experiences. Otherwise the data presented was sound, clearly itemised and presented

Your comment is important and we completely agree with you. However, this is only a qualitative study. There were no open questions and therefore no verbal aspects are included. We will consider this positively in the following studies.

  • The manuscript is clear, relevant for the field and presented in a well-structured manner 
  • The cited references are relevant
  • The manuscript’s results are reproducible and based on the details given in the methods section
  • The figures/tables/images/schemes are appropriate and are easy to interpret and understand
  • The conclusions are consistent with the evidence and arguments presented
  • The ethics statement is included  

Comments on the Quality of English Language

There is some scope for the article to be edited and checked for English grammar and syntax 

We edited all the paper for improved for English grammar and syntax 

Your review was of great value for us

Thank you very much

The authors

Reviewer 2 Report

Comments and Suggestions for Authors

The study is interesting and addresses a relevant issue in teacher education today, linked to models of practice in curricula and their projections in the practice of the profession.  However, the following aspects need to be reviewed:

 1.            Between lines 25 and 27, the following statement "However, there is still a lack of in-depth research on the effect of the collaborative model on novice teachers, who either persevere, or alternatively chose to drop out from the profession", needs to be backed up with the corresponding reference or references, because it is presented as an unsupported value judgement.

2. The hypothesis of the study is not said.

3. The research design needs to be explained more clearly, because there is some doubt as to whether it was an experimental study or a non-experimental comparative (correlational) design, since at times it talks about a control group and an experimental group.

4.            The sample selection procedure needs to be clarified.

5.            It is necessary to explain how the instrument used was validated, its psychometric qualities and the indicators (variables) considered for each of the four aspects measured (see lines 169-177). As the variables measured for each dimension are not specified, there are certain inconsistencies in the presentation of the results.

6.            In line 299, the word "Summary" should be removed, the summary has already been presented in the corresponding section. In this item, only the discussion of the results should be considered, and I recommend separating the discussion from the conclusions.

7.            Revise reference 2. The doi does not correspond with the referenced article (line 375), reference 33. (line 438) has errors. In addition, it is recommended to incorporate the respective doi in the references of the following lines: 389, 397, 400, 420, 450, 454, 456, 458.

Author Response

Dear reviewer,

First, we would like to thank you wholeheartedly for your eye-opening comments. These have been tremendously helpful in guiding the major changes we have made to this paper. We have made significant changes based on all your comments as you can see in "Track the changes". We would like to emphasize the following points that have been substantially modified in line with your comments.

The study is interesting and addresses a relevant issue in teacher education today, linked to models of practice in curricula and their projections in the practice of the profession.  However, the following aspects need to be reviewed:

  1. Between lines 25 and 27, the following statement "However, there is still a lack of in-depth research on the effect of the collaborative model on novice teachers, who either persevere, or alternatively chose to drop out from the profession", needs to be backed up with the corresponding reference or references, because it is presented as an unsupported value judgement.

We added the reference.

  1. The hypothesis of the study is not said.

We added the hypothesis in the methodology section.

  1. The research design needs to be explained more clearly, because there is some doubt as to whether it was an experimental study or a non-experimental comparative (correlational) design, since at times it talks about a control group and an experimental group.

We corrected it according to your suggestions.

  1. The sample selection procedure needs to be clarified.

We added a clarification.

  1. 5.            It is necessary to explain how the instrument used was validated, its psychometric qualities and the indicators (variables) considered for each of the four aspects measured (see lines 169-177). As the variables measured for each dimension are not specified, there are certain inconsistencies in the presentation of the results.

  1. In line 299, the word "Summary" should be removed, the summary has already been presented in the corresponding section. In this item, only the discussion of the results should be considered, and I recommend separating the discussion from the conclusions.

 We corrected this according to your suggestiations.

  1. 7.            Revise reference 2. The doi does not correspond with the referenced article (line 375), reference 33. (line 438) has errors. In addition, it is recommended to incorporate the respective doi in the references of the following lines: 389, 397, 400, 420, 450, 454, 456, 458.

We revised and add most of the references DOI. After many tracking attempts Some of them doesn’t have a DOI.

Your review was of great value for us

Thank you very much

The authors

Round 2

Reviewer 2 Report

Comments and Suggestions for Authors

The work was substantially improved, with some items showing greater theoretical robustness and more clarity in the method used. However, it is still necessary to address some issues that hinder the reading and, so, the new revision, among which I highlight the following:

·         - The information presented on pages 35-43 is repeated on lines 16-20.

·         - Between lines 116-118, the idea is unfinished.

·        - What is said between lines 212 and 228 has already been said, I recommend sticking only to the subtitle under consideration.

·       -  Information is repeated in the method (see lines 330-338; 358-362; 445-450).

When comparing this new document and the original, I have the impression that the use of the "change control" tool is not well achieved, because at times there is doubt about the new information incorporated and that which corresponds to the original manuscript. This is because the new version presents paragraphs in italics in which background information is reiterated, so this form of presentation hinders the evaluation.

Author Response

Dear reviewer.
We have carefully gone through all your comments.

"

The information presented on pages 35-43 is repeated on lines 16-20.

  •        - Between lines 116-118, the idea is unfinished.
  •       - What is said between lines 212 and 228 has already been said, I recommend sticking only to the subtitle under consideration.
  •      -  Information is repeated in the method (see lines 330-338; 358-362; 445-450).

Thank you very much they are important and very valuable.
All duplicate sentences have been removed.
We corrected and updated the sentences in the lines you mentioned
I removed the "track changes" i guess now the clean file will be much clearer and readable.

thank you

the author's